# Convective self–aggregation in a mean flow

Hyunju Jung[1,2], Ann Kristin Naumann[1], and Bjorn Stevens[1]

[1]Max Plank Institute for Meteorology, Hamburg, Germany
[2]Currently at the Institute of Meteorology and Climate Research (IMK–TRO), Karlsruhe Institute of Technology (KIT), Karlsruhe, Germany

**Correspondence:** Hyunju Jung (hyunju.jung@kit.edu)

**Abstract.** Convective self-aggregation is an atmospheric phenomenon seen in numerical simulations in a radiative convective equilibrium framework thought to be informative of some aspects of the behavior of real-world convection in the deep tropics. We impose a background mean wind flow on convection-permitting simulations through the surface flux calculation in an effort to understand how the asymmetry imposed by a mean wind influences the propagation of aggregated structures in convection. The simulations show that with imposing mean flow, the organized convective system propagates in the direction of the flow but slows down compared to what pure advection would suggest, and eventually becomes stationary relative to the surface after 15 simulation days. The termination of the propagation arises from momentum flux, which acts as a drag on the near-surface horizontal wind. In contrast, the thermodynamic response through the wind-induced surface heat exchange feedback is a relatively small effect, which slightly retards the propagation of the convection relative to the mean wind.

## 1 Introduction

In this manuscript we explore the simplest possible configuration that allows the interaction of a convective cluster with a mean flow. This is motivated by a desire to better understand processes influencing the propagation of organized deep convection in the tropics. In simulations of radiative convective equilibrium (RCE), a single aggregated cluster can develop from randomly distributed convective fields despite homogeneous initial conditions, boundary conditions, and forcing (e.g., Tompkins and Craig, 1998; Bretherton et al., 2005; Coppin and Bony, 2015; Hohenegger and Stevens, 2016). Convective self-aggregation exhibits many similarities to organized deep convection in the tropics including phenomena such as the Madden-Julian Oscillation (MJO), which is an eastward-propagating intraseasonal variability in the tropics (Madden and Julian, 1971, 1972). Some studies suggested that the MJO may itself be an expression of self-aggregation (Raymond and Fuchs, 2009; Dias et al., 2017). This idea is supported by recent studies showing that MJO-like phenomena are observed in rotating RCE simulations in cloud-resolving models (Arnold and Randall, 2015; Khairoutdinov and Emanuel, 2018). Further support for this point of view comes from the observational study by Tobin et al. (2013), who found that the mean state of the atmosphere during an active phase of the MJO resembles the self-aggregation state in the sense that a higher degree of the convective

organization is associated with more outgoing longwave radiation. This leads us to the more basic question of how convective

self-aggregation responds to the imposition of a mean flow.

Emanuel (1987) and Neelin et al. (1987) proposed that the interaction between wind and the surface enthalpy flux in a mean flow may be important for the MJO propagation. They demonstrated that in mean easterlies winds are amplified by the convective scale circulation to the east of convection, leading to a positive anomaly of the surface enthalpy flux. This favors the initiation of convection on the upwind side of the cluster, resulting in the upstream propagation of convection. Emanuel

(1987) called this the wind-induced surface heat exchange (WISHE) feedback. Self-aggregation studies also showed that in the absence of mean wind, WISHE contributes to the maintenance of aggregation as the enhanced surface enthalpy flux favors the development of deep convection on the periphery of the existing convection (Bretherton et al., 2005; Wing and Emanuel, 2014; Coppin and Bony, 2015).

This line of thinking leads us to attempt to study a much simpler problem, which is how convective-self aggregation responds

to the imposition of a background mean flow. As a step, we focus on how asymmetries in the surface flux, in response to a mean flow, affect the propagation of a convective cluster in RCE. We impose a large-scale mean flow in simulations of RCE in the form of a shear free wind, a setup that has not been investigated in previous simulations of RCE. We hypothesize that on the upwind side of a convective cluster, the mean flow adds constructively to the near-surface component of the convective scale circulation, enhancing the surface enthalpy flux, and vice versa on the downwind side. The asymmetry in the thermodynamic

response to the mean wind leads to a slow upwind propagation of the deep convective system. In addition to the thermodynamic response, we also investigate the dynamic response to the mean flow, that is how the modified surface wind field affects the surface momentum fluxes. The simulations show that the thermodynamic response to asymmetry in the mean winds is strongly coupled to changes in the momentum budget, which equilibrates the near-surface winds, due to a mean wind contribution to the surface drag in ways that damps and eventually eliminates asymmetries in the surface heat and moisture fluxes. We perform a

mechanism denial experiment to suppress the dynamic response and quantify to what extent the propagation can be attributed to the thermodynamic response.

In Sect. 2 we describe the simulation design including a mechanism denial experiment and discuss the limitations of the setup. Sect. 3 shows how a convective cluster propagates in the mean flow with different mean wind speeds. In Sect. 4 we examine the thermodynamic response. In Sect. 5 we explore the surface momentum flux and discuss the mechanism denial

experiment. Conclusions are given in Sect. 6.

## 2  Simulation setup

We conduct numerical simulations using the University of California Los Angeles Large-Eddy Simulation (UCLA-LES) model. The UCLA-LES solves the anelastic equations with a third-order Runge Kutta method for the temporal discretization and with centered difference in space for momentum (Stevens et al., 2005). Full radiation is computed by using Monte

Carlo spectral integration (Pincus and Stevens, 2009), including radiative properties of ice clouds (Fu and Liou, 1993). A two-

moment microphysical parameterization for mixed-phase clouds is used to represent cloud water, rain water, cloud ice, snow, and graupel, explicitly (Seifert and Beheng, 2006a, b). Sub-grid scale fluxes are modeled with a Smagorinsky model.

A $576 \times 576 \times 27\,\mathrm{km}^3$ domain size is used with horizontal grid spacing of $3\,\mathrm{km}$ to resolve deep convection. The 63 vertical grid levels are stretched, starting from a grid spacing of $75\,\mathrm{m}$ at the first model level up to $1367\,\mathrm{m}$ near the model top. The small vertical grid spacing near the surface allows us to better resolve the boundary layer's vertical structure. There is no rotation and no diurnal cycle. The experimental design of the UCLA-LES simulations follows Hohenegger and Stevens (2016). In contrast to using interactive sea surface temperature (SST) of their experiments, we prescribe an SST of $301\,\mathrm{K}$.

We consider two types of simulations. In a first set of experiments we conduct numerical simulations with different background wind speeds. In an effort to isolate the thermodynamic effects of the convective circulation on the evolution of the self-aggregated convective cluster, we subject the flow to mean wind whose presence is encoded through the surface fluxes. This is equivalent to simulating a situation subject to a large-scale mean wind using a Galilean transform to avoid numerical artifacts of advection (Matheou et al., 2011) but neglecting any restoring force for the wind. Under such a transform, surface fluxes are not invariant, and the effect of the mean wind is accounted for only through the surface flux calculation, which spins down the wind. Effects of WISHE-like asymmetries in the surface fluxes will then be present in so far as they affect the flow on time-scale shorter than those associated with the spin-down of the mean wind due to surface drag. In the long run when the effect of the modified surface fluxes is transferred to the atmosphere above by the momentum flux, the velocity in the atmosphere naturally reduces towards that of the surface, until the whole column is in balance again and stagnant compared to the surface. (Note that this equilibrium response is different from the equilibrium response of a nudging approach, where a background flow is maintained. For the transient response we expect a similar behavior of both approaches.) For the mechanism denial experiment a mean flow over the surface is maintained by including the influence of the mean wind only in the surface enthalpy equation but not in the surface momentum equation. The first set of experiments is described in Sect. 2.1, and the additional experiment in Sect 2.2.

## 2.1  Experiments with a mean wind encoded in the surface fluxes

The surface fluxes, including the momentum flux ($F_{\mathrm{m}}$) at the surface and the surface enthalpy flux ($F_{\mathrm{h}}$), are defined as:

$$
\begin{aligned}
F_{\mathrm{m}} &= \rho\,(\overline{w'u'}^2 + \overline{w'v'}^2)^{\frac{1}{2}}|_{\mathrm{sfc}}, \\
F_{\mathrm{h}} &= \rho\,(c_p\,\overline{w'\theta'} + l_{\mathrm{v}}\,\overline{w'q'})|_{\mathrm{sfc}},
\end{aligned}
\tag{1}
$$

with $\rho$ being the air density at the surface, $c_p$ the isobaric specific heat and $l_{\mathrm{v}}$ the specific enthalpy of vaporization. The covariances $\rho\overline{w'u'}$ and $\rho\overline{w'v'}$ represent the $x$- and $y$-component of momentum fluxes in kinematic units, respectively. The terms $\overline{w'\theta'}$ and $\overline{w'q'}$ represent the near-surface turbulent fluxes of potential temperature and specific humidity, respectively. The turbulent fluxes are calculated from the turbulence scales of velocity $u_*$, temperature $\theta_*$ and humidity $q_*$ as $\overline{w'u'}^2 + \overline{w'v'}^2 = -u_*^2$, $\overline{w'\theta'} = -u_*\theta_*$ and $\overline{w'q'} = -u_*q_*$. The scale values are computed from profiles of horizontal velocity, temperature and humidity in the boundary layer based on similarity functions ($\Psi_m$, $\Psi_h$) proposed by Dyer and Hicks (1970), Businger (1973),

and Dyer (1974). In the model, $u_*$ is proportional to the near-surface horizontal wind $u_h$ which is defined as the wind at the first level above the surface, which is at $37.5\,m$ in our case. We modify $u_h$ by adding a mean flow $u_b$ to it:

$$u_h = \sqrt{(u + u_b)^2 + v^2}. \qquad (2)$$

The modification makes the model see the $x$-component wind of $u + u_b$ in the surface flux formulation. Physically, this is
equivalent to the Galilean transform that works as if we move the surface with a velocity of $-u_b$, so that it is analogous to putting the atmospheric system on a conveyor belt. From the point of view of an observer fixed relative to the moving surface, at $t = 0$ the air velocity at all levels is $u_b$, and the model framework is also moving at speed $u_b$. This surface flux modification allows us to have a shear free mean flow in the simulations. In retrospect, this modification ends up being effective only to a limited extent, as the advantage of a Galilean transformation to avoid numerical errors from advection is lost when the air and
the convective cluster start to move through the grid boxes. For future studies that aim to study the interaction of convective self-aggregation with a mean flow, mechanisms for maintaining the mean flow must be included (e.g. a nudging of a large-scale flow), which couples the thermodynamic questions we had wished to study to dynamical ones.

The aggregated state in simulations of RCE reveals hysteresis; it hardly returns to the random occurrence of convection once an aggregated state is established (Khairoutdinov and Emanuel, 2010; Muller and Held, 2012). We start from an aggregated
state in order to separate the effect of a mean wind on the evolution of self-aggregation from its initiation. For this purpose, we run a simulation without a mean wind for 26 days until the convection is fully aggregated. The time scale of self-aggregation in our simulations is comparable to other self-aggregation studies in a square domain (Wing and Emanuel, 2014; Holloway et al., 2017; Arnold and Putman, 2018). We then restart the simulations from the aggregated state, but with a mean wind imposed. The specification of the surface fluxes are described above. Each experiment with $u_b$ ranging from 0 to $4\,m\,s^{-1}$ is simulated for
additional 20 days. Organized convection disaggregates when $u_b$ is stronger than $4\,m\,s^{-1}$. Since disaggregation of organized convection is not the focus of this study, the experiments for $u_b$ of 0, 2 and $4\,m\,s^{-1}$ are discussed and will be denoted by UB0, UB2 and UB4, respectively.

## 2.2    Mechanism denial experiment

UB0, UB2 and UB4 indicate that the dynamic feedback significantly modulates the propagation of the convective system, as
the surface momentum flux $F_m$ interacts with the near-surface wind $u_h$ through the velocity scale $u_*$ (Sect. 2.1). To isolate the role of the thermodynamic feedback, we perform a mechanism denial experiment wherein we suppress the influence of $F_m$ on $u_h$. The surface fluxes are determined by the turbulent fluxes at the surface (Eq. 1), and the turbulent fluxes are obtained from the turbulence scales: $u_*$, $\theta_*$ and $q_*$ (Sect. 2.1). We disable the effect of the surface momentum flux by setting $u_*$ to a constant value for the computation of $\overline{w'u'}$ and $\overline{w'v'}$ (thus, $F_m$), but using the modeled $u_*$ for computation of $\overline{w'\theta'}$ and $\overline{w'q'}$ (thus, $F_h$)
as in UB0, UB2 and UB4. For the momentum flux, we prescribe $u_*$ as a constant value of $0.09\,m\,s^{-1}$ obtained by averaging $u_*$ over the simulation domain and the last 20 simulation days in UB0. For the mechanism denial experiment, $u_*$ is temporally

and spatially constant to disable the dynamic feedback, but remains variable for the surface enthalpy flux in order to retain the WISHE feedback.

In UB0 convection begins to be organized into a single cluster at around day 22, so we restart a simulation with an uncoupled $F_m$ but without mean wind from day 22 in order to confirm that the suppression of the dynamic feedback does not affect the aggregation. The simulation with the uncoupled $F_m$ from day 22 maintains convective self-aggregation towards the end of the simulation period (day 46), and the horizontal scale of the convective cluster in this simulation is approximately $100\,\mathrm{km}$, which is comparable to that in UB0 (not shown). In the same way as the experiments with coupled $F_m$ (Sect. 2.1), $u_b$ of $2\,\mathrm{m\,s^{-1}}$ is imposed on the the simulation with the uncoupled $F_m$ after day 26. The experiment with uncoupled $F_m$ will be denoted by UB2_unius.

For the remainder of the study we refer to the simulation day, where we begin to impose the background wind, as day 0 (day 26 above). For example, the time when we restart the denial experiment without mean wind (day 22) would be equivalent to day $-4$, and the time when the mean wind is introduced to be imposed to the denial experiment (day 26) is day 0 from now on.

## 3    Propagation speed of the organized convective cluster

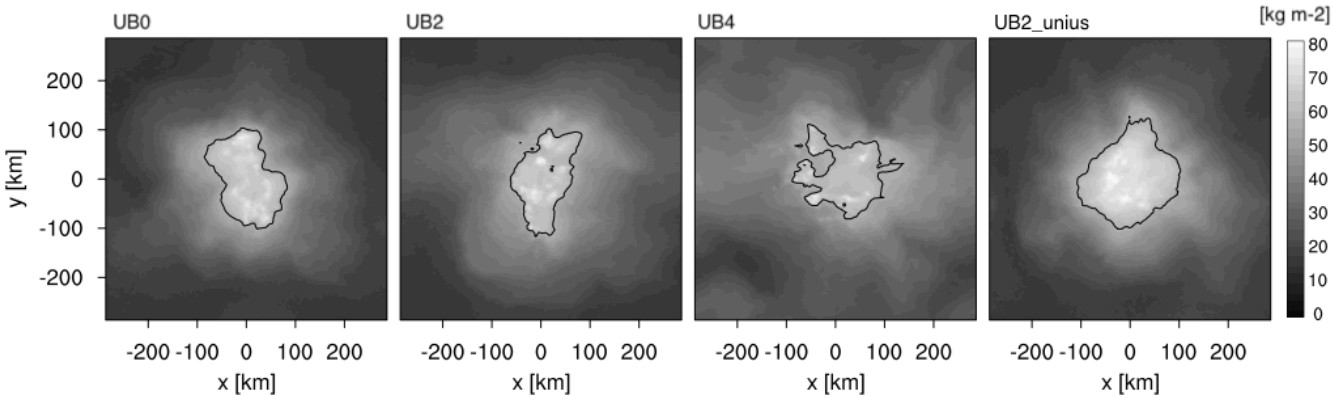

**Figure 1.** Daily average precipitable water on day 19. Black contours indicate where precipitable water is equal to $58\,\mathrm{kg\,m^{-2}}$.

Figure 1 illustrates the daily average spatial pattern of the convective cluster on the last day in the experiments. All simulations show that the quasi-circular pattern of the convective cluster lasts until the end of the simulation period, and the horizontal scale of the cluster size is comparable among all simulations, although the spatial variability of precipitable water is weak for UB4 compared to the other experiments. The standard deviation of the daily average precipitable water on the last simulation day is 14.2, 12.1 and $10.4\,\mathrm{kg\,m^{-2}}$ for UB0, UB2 and UB4, respectively. This standard deviation varies in time and, e.g., is as low as $10.9\,\mathrm{kg\,m^{-2}}$ on day 6 in the control case UB0. The domain mean precipitable water on the last day increases with increasing $u_b$, having the daily mean value of 26.5, 30.4 and $34.3\,\mathrm{kg\,m^{-2}}$ for UB0, UB2 and UB4. The larger domain mean

precipitable water with increasing $u_b$ might be associated with our simulation setup of a propagating cluster in double periodic boundary condition which results in nine full transits through the domain in case of UB4 (Matheou et al., 2011). Despite this artifact, the convective cluster remains organized over the simulation period in all experiments and we expect this difference to play a minor role in the following analysis.

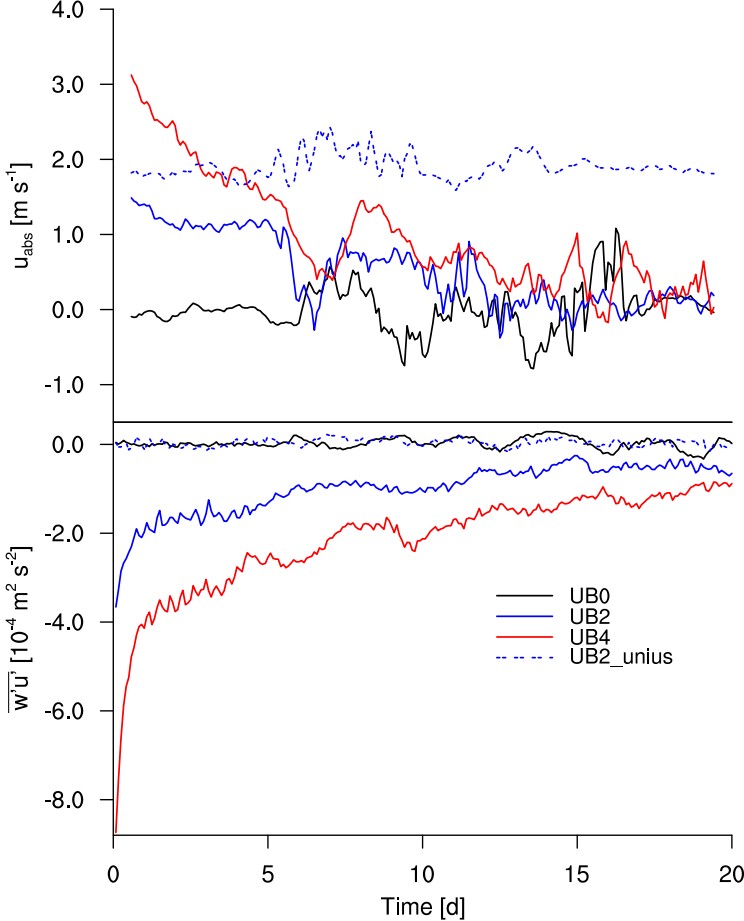

**Figure 2.** Temporal evolution of (top) $u_{abs}$ in the $x$-direction and (bottom) domain-averaged $\overline{w'u'}$ at the surface. Day 0 corresponds to the day when $u_b$ begins to be imposed.

We estimate the propagation speed of a convective cluster by tracking the cluster in the simulation domain. We find all grid columns where the precipitable water (PW) is greater than $62\,\mathrm{kg\,m^{-2}}$, and define a convective cluster with the grid points at each output time step. The motion of the cluster is determined by tracking the PW-weighted mean center of the cluster with time. Only $x$-direction motion is considered because the cluster propagates in the $x$-direction. Changing the threshold level does not affect the estimated propagation speed. Since in the model setup the surface effectively moves with a constant

speed below the atmospheric column, the absolute propagation velocity of the convective cluster to the model surface $u_{\mathrm{abs}}$ is calculated as the sum of the relative velocity of the cluster to the model grid $u_{\mathrm{rel}}$ and the mean wind speed $u_{\mathrm{b}}$:

$$u_{\mathrm{abs}} = u_{\mathrm{rel}} + u_{\mathrm{b}}. \tag{3}$$

When $u_{\mathrm{rel}} = 0\,\mathrm{m\,s^{-1}}$, the convective cluster remains motionless in the model reference frame but is effectively moving at the speed of $u_{\mathrm{b}}$ by virtue of the Galilean transformation (pure advection). If the hypotheticated effect of WISHE was realized then the convective cluster would move against the mean wind (e.g., $u_{\mathrm{rel}} < 0\,\mathrm{m\,s^{-1}}$). Thus, we expect $u_{\mathrm{abs}} < u_{\mathrm{b}}$ if the WISHE feedback regulates the propagation of the convective cluster.

Figure 2 (top) shows $u_{\mathrm{abs}}$ for each experiment. A 24-hour running average is applied to the temporal evolution of $u_{\mathrm{abs}}$ to present the long-term evolution more clearly. After imposing $u_{\mathrm{b}}$, the convective cluster begins to propagate. For the simulations where the momentum fluxes are allowed to feel the effect of the mean wind, $u_{\mathrm{abs}}$ decreases from what pure advection would suggest to near-zero values at day 15. The decrease of $u_{\mathrm{abs}}$ corresponds to our hypothesis ($u_{\mathrm{abs}} < u_{\mathrm{b}}$), but is masked by the spin-down of the mean wind due to surface drag. Estimating the final value of $u_{\mathrm{abs}}$ by averaging it over the last five days, we arrive at $0.23 \pm 0.31$, $0.10 \pm 0.47$ and $0.29 \pm 0.76\,\mathrm{m\,s^{-1}}$ for UB0, UB2 and UB4, respectively. (At this point the convective cluster appears stationary to the observer.) Additional simulations with $u_{\mathrm{b}}$ of 1 and $3\,\mathrm{m\,s^{-1}}$ show agreement in that the propagation speed decreases in the first few days and eventually the propagation speed converges to zero (not shown). Additional simulation days for UB4 (until day 30) corroborate that UB4 reaches a quasi-equilibrium state (not shown). The strong fluctuation around the mean is due to the oscillating features of aggregation (Bretherton et al., 2005; Windmiller and Hohenegger, 2019; Patrizio and Randall, 2019). This fluctuation hinders our ability to unambiguously distinguish between a slow propagation speed and a stationary one, although its amplitude is comparable to the one with no mean wind (UB0). Since the cluster is formed by a group of individual convective cells, the shape of cluster is not firmly fixed. The cluster expands and contracts in time (though not necessarily in all directions at the same time, see the daily PW for UB2 in Fig. 1) and sometimes smaller convective cells emerge outside the main cluster (see the cloud top height for UB0 in Fig. 6). Qualitatively the simulations indicate that the aggregated cluster initially moves with the wind. As the simulations with the mean winds proceed the convective clusters develop into the wind and as the mean wind spins down they become stationary with respect to the surface.

In the following sections we examine if the tendency towards stationarity is a consequence of WISHE-like symmetries by means of an upstream/downstream difference.

## 4   Thermodynamic process

The temporal evolution of the propagation speed demonstrates that the spin-down of the propagation speed occurs over a week whose time scale is longer than the convective adjustment time scale, which is in the order of hours, and the convective cluster settles around two weeks after it begins to propagate. We focus on two simulation periods: the transient phase for the first five

days (day 0-4) when $u_{\mathrm{abs}}$ prominently decreases and compare it to the quasi-stationary stage for the last five days (day 15-19) when $u_{\mathrm{abs}}$ is near-zero. Quantities are averaged over these periods.

The surface enthalpy flux is larger on the upwind side of a convective cluster than on the downwind side through WISHE, i.e., the modulation of $u_{\mathrm{abs}}$. Convection is expected to locate over the maximum boundary layer equivalent potential temperature $\theta_{\mathrm{e}}$. Hence to understand how WISHE affects its distribution we calculate the flux of $\theta_{\mathrm{e}}$ approximately as $\overline{w'\theta_{\mathrm{e}}'} \approx \overline{w'\theta'} + \frac{l_{\mathrm{v}}}{c_p}\left(\frac{p_0}{p}\right)^{\frac{R_{c_p}}{c_p}} \overline{w'q'}$. Its form is analogous to the enthalpy (or moist static energy) flux. Focusing on the budget of $\theta_{\mathrm{e}}$ allows us to investigate whether the development of convection is associated with the positive anomaly of the surface enthalpy flux.

Figure 3 (top) illustrates how $\overline{w'\theta_{\mathrm{e}}'}$ varies from the center of the convective cluster ($r = 0\,\mathrm{km}$) into the environment surrounding the cluster. We place the center of the convective cluster in the center of the domain at each output time step, average the physical quantities, and partition the domain diagonally into quarters, thus defining an upwind area, a downwind area and crosswind areas. Only the upwind and downwind areas are illustrated. The distribution of $\overline{w'\theta_{\mathrm{e}}'}$ for UB0 indicates that the surface enthalpy flux is strengthened because the low-level convergence of the convective circulation intensifies the near-surface horizontal wind in the vicinity of the main convective cluster which is also observed in other RCE studies (e.g., Bretherton et al., 2005; Coppin and Bony, 2015). As we expected, for UB2 and UB4 in the transient phase $\overline{w'\theta_{\mathrm{e}}'}$ is enhanced on the upwind side and suppressed on the downwind side. These enhancement and suppression of $\overline{w'\theta_{\mathrm{e}}'}$ become stronger with increasing $u_{\mathrm{b}}$. In the quasi-stationary stage the spatial distribution of $\overline{w'\theta_{\mathrm{e}}'}$ becomes symmetric.

In the model, the surface enthalpy flux is determined by the difference between the wind speed near the surface and the velocity of the surface, which is equal to $0\,\mathrm{m\,s^{-1}}$, as well as the vertical differences of specific humidity and potential temperature between the surface and the first level above the surface. The vertical differences of humidity and temperature do not have significant asymmetric features, but $u_{\mathrm{h}}$ shows the same transition from asymmetry to symmetry over time as seen in $\overline{w'\theta_{\mathrm{e}}'}$ (Fig. 4). Immediately after $u_{\mathrm{b}}$ is imposed, $u_{\mathrm{h}}$ is intensified on the upwind side and reduced on the downwind side as one would expect from a superposition of $u_{\mathrm{b}}$ and the local circulation associated with the convective cluster. In the later stage of imposing $u_{\mathrm{b}}$, the drag has transported its signal through the near-surface layers and $u_{\mathrm{h}}$ attains a comparable magnitude of wind speed on the upwind and downwind sides. For UB4, the off-centered local minimum of $u_{\mathrm{h}}$ around $r = 0\,\mathrm{km}$ is due to the strong modeled wind $u$ on the downwind side in the opposite direction to $u_{\mathrm{b}}$. The distribution of $u_{\mathrm{h}}$ indicates that the adjustment of the near-surface wind field modifies the response of the convection to the mean wind that one would expect from thermodynamic consideration alone.

## 5  Dynamic process

Without Coriolis force, the tendency of the horizontal wind is obtained as follows:

$$
\begin{aligned}
\frac{\partial u}{\partial t} &= -\mathbf{V}\cdot\nabla u - c_p\theta\frac{\partial \pi}{\partial x} + \frac{1}{\rho}\frac{\partial \rho \overline{w'u'}}{\partial z}, \\
\frac{\partial v}{\partial t} &= -\mathbf{V}\cdot\nabla v - c_p\theta\frac{\partial \pi}{\partial y} + \frac{1}{\rho}\frac{\partial \rho \overline{w'v'}}{\partial z},
\end{aligned}
$$

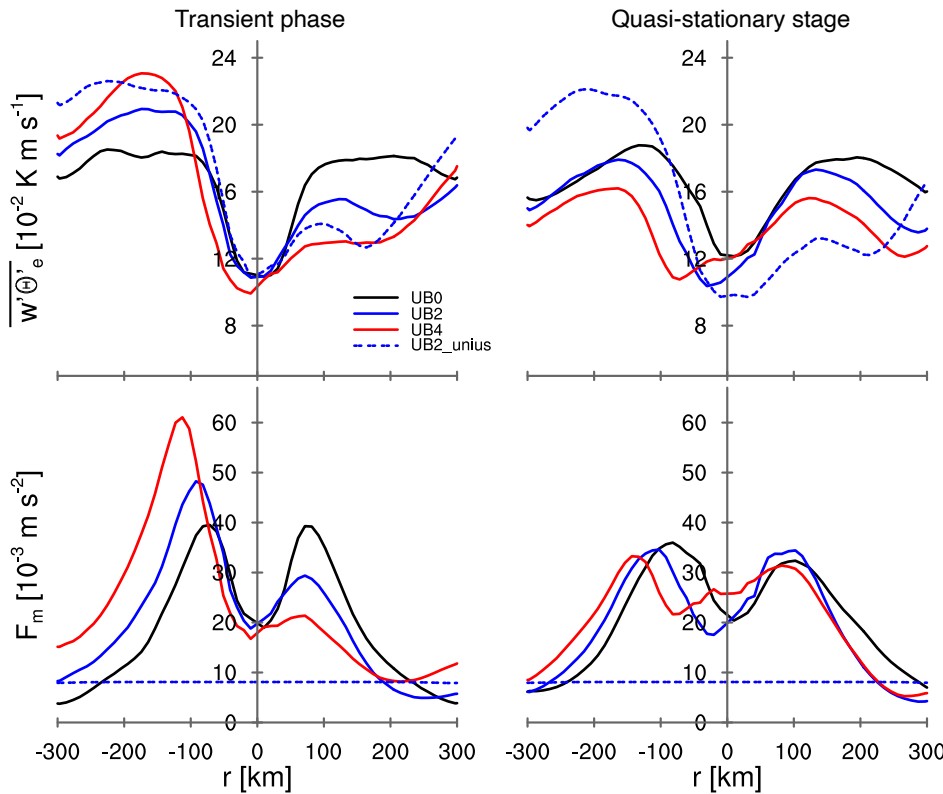

**Figure 3.** Radial distributions of the azimuthally averaged (top) $\overline{w'\theta'_e}$ and (bottom) $F_m$. Quantities are averaged over 5 days and $10\,\mathrm{km}$ in $r$-direction. The averaged quantities for (left) transient stage over day 0 to 4 and (right) quasi-stationary stage over 15 to 19 are illustrated. The negative and positive values of $r$ represent the upwind area and downwind area, respectively.

with $\mathbf{V}$ the vector wind, $\mathbf{V} = (u, v, w)$. The first term on the right-hand side represents the advection and the second term represents the pressure gradient force with the Exner function $\pi = \left(\frac{p}{p_0}\right)^{\frac{R_d}{c_p}}$. The third term on the right-hand side represents the contribution of friction to the wind tendency and is related to $F_m$ (Eq. 1). For UB2 and UB4 the vertical profile of the $x$-component of the wind in the quasi-stationary stage differs from the initially prescribed shear-free profile, while remaining constant with height for UB0 and UB2_unius (Fig. 5 left). When $u_b$ interacts with $F_m$, the surface drag transports its signal

through the atmosphere and the horizontal wind is substantially slowed down, particularly near the surface. The convective cluster is moving with the lower-tropospheric flow well before the whole tropospheric momentum is balanced. In the long term, we expect a balance to ensue with the whole column resting compared to the surface in UB2 and UB4.

  As seen in $\overline{w'\theta'_e}$ and $u_h$, the spatial distribution of $F_m$ shows an asymmetry with respect to the center of the convective cluster in the transient phase and a symmetry in the quasi-stationary stage (Fig. 3 bottom). A larger $F_m$ corresponds to a stronger drag

on $u_h$. As a result of the intensified $u_h$, the enhanced $F_m$ on the upwind side exerts a strong drag on $u_h$ in the transient phase, and consequently, reduces $u_h$ on the upwind side in the quasi-stationary stage. In contrast, the suppressed $F_m$ on the downwind

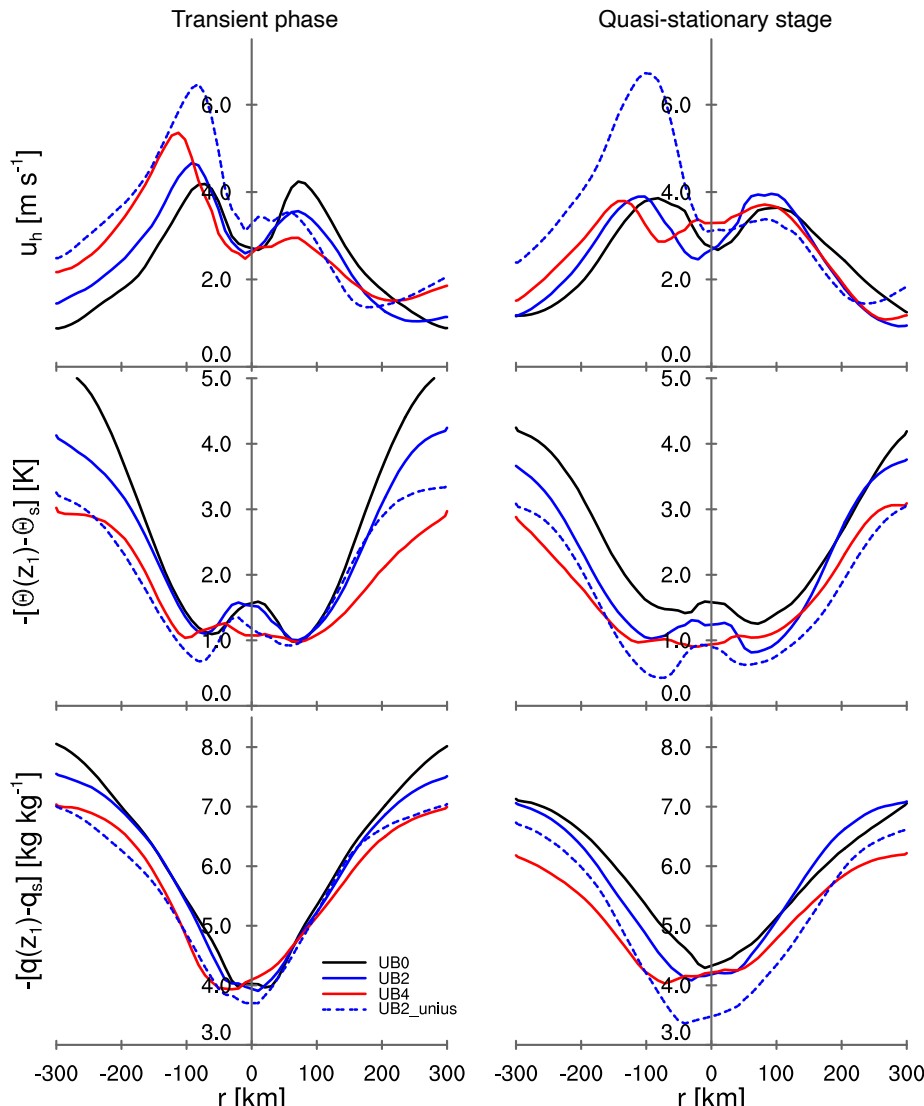

**Figure 4.** As in Fig. 3, but for (top) the near-surface horizontal wind $u_h$, (middle) the vertical difference of potential temperature $-[\theta(z_1) - \theta_s]$, and (bottom) the vertical difference of humidity $-[q(z_1) - q_s]$. The subscription $s$ denotes the property at the surface and $z_1$ represents the first model level above the surface, which is at $37.5\,\mathrm{m}$ in our simulations.

side generates a weak drag, allowing $u_h$ on the downwind side to become stronger in the quasi-stationary stage. This difference, or asymmetry, in the drag acts as a source of momentum that accelerates the mean wind until it balances the mean wind, thereby eliminating the asymmetry in the drag by symmetrizing $\overline{u_h}$. As a result, the symmetric $\overline{u_h}$ in the quasi-stationary stage affects not only the spatial distribution of $F_m$ but also that of $\overline{w'\theta'_e}$. The upstream/downstream difference cannot be sustained close to

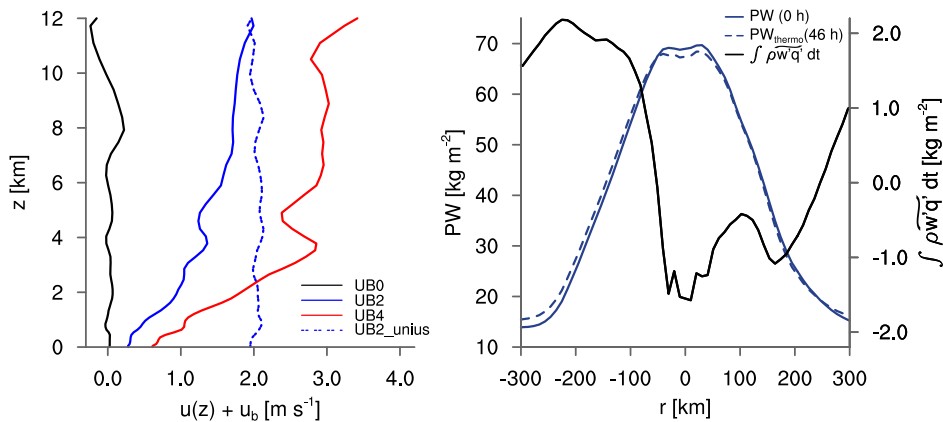

**Figure 5.** (Left) vertical profile of the domain-mean $x$-component wind as sum of the modeled wind $u(z)$ in the $x$-direction and $u_\mathrm{b}$ for the quasi-stationary stage. Note that the horizontal wind considers the Galilean transformation by including $u_\mathrm{b}$. (Right) radial distributions of PW at $0\,\mathrm{h}$, the estimated PW at $46\,\mathrm{h}$ due to the thermodynamic process alone, and the accumulated surface moisture flux anomaly from $0\,\mathrm{h}$ to $46\,\mathrm{h}$. The quantities are azimuthally averaged.

the surface because the momentum exchange limits it in our simulations. This does not rule out a sustained effect in a different system where there is an active dynamical driving of a low-level flow.

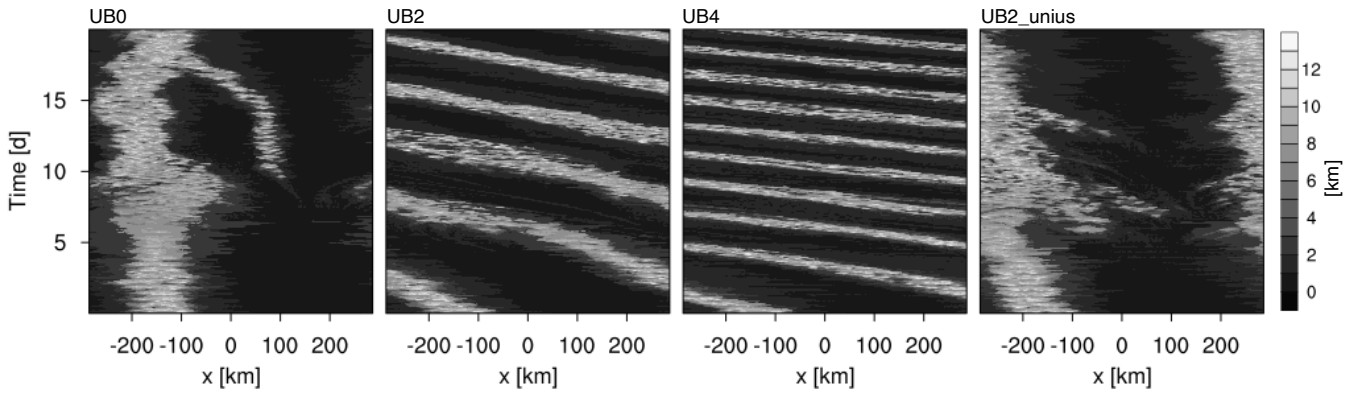

**Figure 6.** Hovmöller diagram of the cloud top height averaged over the $y$-axis for each experiment. This displays the cloud top movement with respect to the model grid, thus $u_\mathrm{rel}$ in Eq. 3.

To isolate the role of a sustained thermodynamic feedback, we perform an additional simulation where $u_*$ is kept constant in space and time for the calculation of $F_\mathrm{m}$ but remains interactive for $\overline{w'\theta'}$ and $\overline{w'q'}$ based on the similarity functions and we use $u_\mathrm{b} = 2\,\mathrm{m\,s^{-1}}$ for the suppressed $F_\mathrm{m}$ experiment (Sect. 2.2). Due to the constant value of $u_*$, the domain-averaged $\overline{w'u'}$ lingers close to zero with small fluctuations for the simulation with suppressed dynamic feedback, UB2_unius, while being


negative immediately after imposing $u_b$ for UB2 (Fig. 2 bottom). The suppression of the dynamic feedback enables $u_h$ to remain asymmetric, and to show stronger maxima in $u_h$ for UB2_unius than for UB2 (Fig. 4 top) and a persistent asymmetry of the surface enthalpy flux (Fig. 3 top). The long-lasting asymmetric feature does not considerably decrease the propagation

speed, resulting in the final value of $u_{abs}$ of $1.88 \pm 0.16 \, \mathrm{m\,s^{-1}}$ for UB2_unius, hence propagating with a velocity only slightly slower than the mean wind speed of $2 \, \mathrm{m\,s^{-1}}$. A Hovmöller diagram of the cloud top height confirms the estimated propagation speed, showing that the convective cluster indeed moves against $u_b$ with a very small value of $u_{rel}$ (Fig. 6). The propagation speed is only about $5\,\%$ smaller than $u_b$ of $2 \, \mathrm{m\,s^{-1}}$, suggesting that this small difference between $u_{abs}$ and $u_b$ can be associated with the thermodynamic feedback alone.

As the surface momentum flux is uncoupled from the near-surface wind field, the displacement of the convective cluster with time can be considered to be a result of the pure thermodynamic process. Assuming that the change of the lateral transport of the moisture flux is negligible, the spatial distribution of PW due to the pure thermodynamic process at a certain time $\mathrm{PW_{thermo}}(t_1)$ is obtained by adding the surface moisture flux anomaly $\widetilde{\rho w'q'}$ integrated over a time period $[t_0, t_1]$ to the initial PW at $t_0$:

$$\mathrm{PW_{thermo}}(t_1) = \mathrm{PW}(t_0) + \int_{t_0}^{t_1} \widetilde{\rho w'q'} \, dt.$$

This simple thermodynamic argument gives us a displacement of $\mathrm{PW_{thermo}}(46\,\mathrm{h})$ from $\mathrm{PW}(0\,\mathrm{h})$ of approximately 10 km (Fig. 5 right), which corresponds to $u_{rel} = -0.06 \, \mathrm{m\,s^{-1}}$ and therefore $u_{abs} = 1.94 \, \mathrm{m\,s^{-1}}$. The estimated displacement of the precipitable water within the given time step due to the moisture flux anomaly agrees well with the estimated propagation speed of $1.88 \pm 0.16 \, \mathrm{m\,s^{-1}}$ for UB2_unius (Fig. 2 top) and confirms that the thermodynamic contribution to the propagation speed of a convective cluster is small in our simulations.

**6    Conclusions**

This study uses a highly simplified framework to understand how the imposition of a mean flow may influence the propagation of organized deep convection. For the simulations, we applied an RCE framework with a horizontal grid spacing of 3 km, with no rotation, and with a prescribed SST of $301\,\mathrm{K}$. We hypothesize that the convective cluster propagates against the mean flow through the WISHE feedback, providing a favorable environment to develop convection on the upwind side of the cluster

(Fig. 7 left). Our idealized simulations with the mean flow exhibit that organized deep convection initially propagates much slower than what pure advection suggests and eventually becomes stationary towards the end of the simulation period regardless of the imposed wind speed. The near-surface wind field in response to the mean flow modifies the surface enthalpy flux and the surface momentum flux. In return, the surface momentum flux acting as a drag decreases the near-surface wind on the upwind side of the convective cluster, and increases it on the downwind side. Because of the surface drag acting on the mean

background wind, the mean momentum near the surface is depleted, and on a timescale of a week the surface relative winds and the surface-relative motion of the convective cluster vanishes (Fig. 7 right). In the simulation with the dynamic feedback

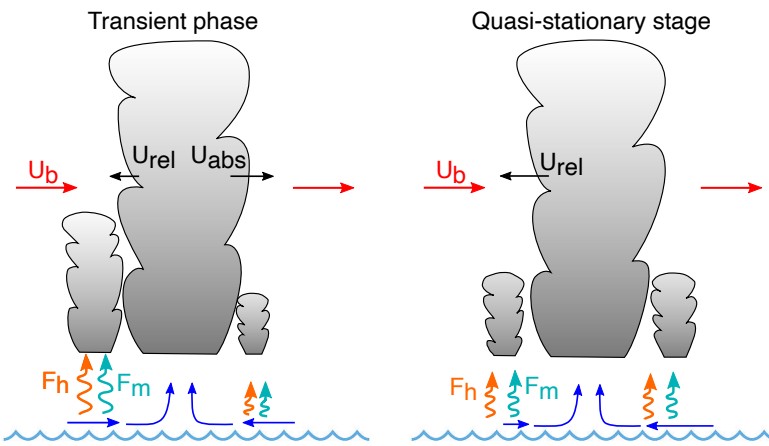

**Figure 7.** Sketch of the convective cluster, the surface wind field, the imposed mean wind ($u_{\mathrm{b}}$), the surface enthalpy flux ($F_{\mathrm{h}}$) and the momentum flux ($F_{\mathrm{m}}$).

removed and the WISHE-induced asymmetry in surface fluxes preserved, the effect on the propagation of convective clusters is small.

While the problem we study is probably too simple to meaningfully inform our understanding of much more complex and larger scale processes like the MJO, it does highlight how a consideration of surface thermodynamic fluxes alone has only a small influence on the propagation of the convective cluster, and how considering these fluxes in isolation of the associated fluxes of momentum, distorts our understanding of the response to the asymmetry imposed by the mean winds. The periodic boundary conditions are limitations of our study, as they cause the effect of anomalously small fluxes to affect the inflow of the region with anomalously large fluxes in ways that damp the effect of the latter. To the extent that WISHE is important for the propagation of convective self-aggregated systems, it would favor large-scale, or solitary systems, so that the moistening that leads the disturbed phase does more than simply offset the drying that lags.

A Galilean transformation can have the advantage of avoiding numerical artifacts of advection. The benefit of the approach, however, ends up being true only to a limited extent, as the convective system starts to propagate through the model grid in our study. Nevertheless, the simulations show that the convective system maintains its thermodynamic structure until the end of the simulation period when $u_{\mathrm{b}} \leq 4\,\mathrm{m\,s}^{-1}$. For future studies, we recommend considering the momentum flux response to a large-scale motion by including a physical mechanism for maintaining a mean flow.

The simplicity of our framework and the difficulties encounter in the setup of the simulations prevent direct inferences from our study for real-world propagating deep convection, let alone the MJO. Compared to typical wind speeds in the tropics, the prescribed large-scale wind speed of up to $4\,\mathrm{m\,s}^{-1}$ in this study is on the low end of the range. Also, feedbacks between the degree of organization and stronger wind speeds remain an open question. Nonetheless, the basic questions it highlights — such as the role of surface momentum fluxes in WISHE-like mechanisms — are likely to be fruitful avenues to explore when pursuing understanding of more complex phenomena.

*Data availability.* The source code of UCLA-LES is released under the GNU General Public License and is publicly available on github (https://github.com/uclales/). The particular version used here is available on request from the authors.

*Author contributions.* BS and AKN developed the idea, designed the experimental setups, and performed initial experiments. HJ analyzed the outputs, performed further experiments, designed and carried out the denial experiment, and interpreted the results together with AKN and BS. HJ prepared the manuscript with contributions from AKN and BS.

*Competing interests.* The authors declare that they have no conflict of interest.

*Acknowledgements.* We thank Dr. Cathy Hohenegger and Dr. Julia Windmiller for helpful discussions of the study. We thank Dr. Martin
Singh for suggesting the analogy of the conveyor belt, and Dr. Tobias Becker and Dr. Caroline Muller for fruitful comments on an early version of the manuscript. We would like to thank two anonymous reviewers and the editor for insightful comments and suggestions on the manuscript. A. K. N. was supported by the Hans-Ertel Centre for Weather Research. This research network of universities, research institutes, and the Deutscher Wetterdienst is funded by the Federal Ministry of Transport and Digital Infrastructure (BMVI). Primary data and scripts used in the analysis and other supplementary information that may be useful in reproducing the author's work are archived by the Max Planck
Institute for Meteorology and can be obtained by contacting publications@mpimet.mpg.de.

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
