# Peer review of "Convective self-aggregation in a mean flow"

_Atmospheric Chemistry and Physics, 2020_

## Referee Comment (RC1) · Anonymous Referee #1 · 24 Oct 2020

This study evaluates the impact of mean flow on the evolution of self-aggregated convection through the imposed wind in surface fluxes. Due to the enhanced surface momentum fluxes, the convection eventually becomes quasi-stationary against the mean wind. The authors further point out that WISHE effects have a relatively small role for the convection to be quasi-stationary. The paper is compact and well written and deserved to be published after addressing the following comments.

Major comments:

1. My major concern is on the experiment design. I think the authors need to emphasize the difference between imposing a mean flow in the cloud-resolving model (i.e., nudge domain averaged horizontal wind in the model) and putting the model domain on a conveyor belt by adding wind only to surface fluxes. The approach proposed in

this study potentially eliminates important processes, including the build-up of near-surface wind shear, the interactions between the mean flow and the cold pools, and the importance of boundary layer processes to the aggregation.

2. The discussion focus on the surface fluxes, but the convection structure change from UB0 to UB4 is not discussed. The only clue given in this manuscript is the cloud top height in Fig.5. Based on my eye measurement, the horizontal scale of the convection in UB0 is around 100 km. But the scale shrinks to 50 km in UB2 and maybe 20 km in UB4 at the end of the simulation. It raises an interesting question: Can we still call convection in UB2 and UB4 aggregated convection (i.e., convection sustained by its circulation). The change in convective structure might also explain why further increase the wind speed, the convection disaggregated.

Minor comments:

In Fig. 1 The Uabs evolution for UB0, UB2, and UB4 are quite different during t=0 to t=5 days. The UB0, which should be very close to no imposed wind, has no fluctuation within first 5 days but fluctuate strongly afterward. This is similar to UB2 but with higher Uabs; on the other hand, the Uabs in UB4 decreases linearly within the first 5 days suggest the active WISHE feedbacks proposed by the authors. Do you have any explanation for this?

Fig. 2 Quantities are averaged over 5 days and 10 km. What does 10 km mean?

Line 48 a grid spacing of 75m at the first model level. Add up to XXX m near the model top.

Line 89 day 22 doesn't make sense here. Do you mean 22 days after restart or 2 days after restart?

Line 91 same problem as line 89.

Fig 2 and others, it's better to put figure legend to all the figures.

---

## Referee Comment (RC2) · Anonymous Referee #2 · 26 Oct 2020

General Comments: This work uses the framework of a radiative convective equilibrium (RCE) experiment to investigate the effect of the wind-induced surface heat exchange (WISHE) mechanism on the moving speed of the active region of convection. By this unique approach, the authors found that the effect of WISHE to slow down the movement is very weak. Specifically, the quantification that the deceleration effect of WISHE is about 5% is valuable. The idea of the Galilean transformation to avoid difficulty in the discretization of the nonlinear term is interesting. Unfortunately, however, I cannot help recommending REJECTION of this work due to the inappropriate experimental configuration at this time. I am wondering if the authors really intended to include the momentum exchange between the surface and the atmosphere. As shown in this work, the surface drag acts to force the atmosphere to move with the ground, and the distinc-

tion between the transient response and the quasi-steady response becomes unclear. In my understanding, there is no positive reason to analyze the transient response. It seems that the authors already noticed this problem, and thus, they had performed the UB2_unius experiment. Would it be so difficult to include influence of the mean wind only in the surface enthalpy equation, but not in the surface momentum equation? One way may be to use two independent $u\_h$ values: one is the same as eq. (2) and the other is without $u\_b$. If the coding is complicating and/or the computational cost is huge, analyzing the transition stage of the system may be acceptable. I think, however, that this is not the case. It would be desirable to revise the experimental settings and redo the whole experiments. After that improvements, the authors' arguments will be clearer, and it will make an essential contribution to the area of the RCE research.

Other Comments: 1. Abstract: "phenomenon found in..." may be replaced with "phenomenon seen in..." 2. It is desirable to review more specifically about the resemblance between the self-aggregation in RCE and the MJO. Since the pure RCE lacks the vertical wind shear, the resemblance between the MJO and the RCE is questionable. 3. Note the number of vertical levels in section 2.

---

## Author Comment (AC1) · 21 Jan 2021

Dear Referee 1,

we thank the reviewer for the valuable comments and suggestions. We have carefully taken into account the comments and have reflected most of the suggestions in the revised manuscript. Here we repeat your comments in bold and write our responses in normal font.

**This study evaluates the impact of mean flow on the evolution of self-aggregated convection through the imposed wind in surface fluxes. Due to the enhanced surface momentum fluxes, the convection eventually becomes quasi-stationary against the mean wind. The authors further point out that WISHE effects have a relatively small role for the convection to be quasi-stationary. The paper is compact and well written and deserved to be published after addressing the following comments.**

**Major comments:**
**1. My major concern is on the experiment design. I think the authors need to emphasize the difference between imposing a mean flow in the cloud-resolving model (i.e., nudge domain averaged horizontal wind in the model) and putting the model domain on a conveyor belt by adding wind only to surface fluxes. The approach proposed in this study potentially eliminates important processes, including the build-up of near-surface wind shear, the interactions between the mean flow and the cold pools, and the importance of boundary layer processes to the aggregation.**

We think that a nudging approach that the reviewer suggests would show a different response in the quasi-equilibrium stage in the way that the background wind is maintained, although we expect the response in the transient phase to be similar between the approach we have taken and the nudging approach.

In the initial stage our approach allows for boundary layer processes like near-surface wind shear (which can still be seen in the quasi-equilibrium stage in Fig. 5 in the submitted manuscript) and the interaction between the mean flow and cold pools. The equilibrium response of the system, when the whole atmospheric column and with it the convective cluster becomes stagnant compared to the surface, i.e. Uabs=0 m/s, is equal to an RCE without a mean flow. Even then we still see local near-surface wind shear and cold pool dynamics but without a mean flow. In response to this comment and also the other reviewer's main comment, we
a) describe the setup more clearly including its implications for boundary layer processes (line 65-68 in the revised manuscript):
    "In the long run with a mean flow the surface transports its signal through the atmosphere, until the whole column is in balance again and stagnant compared to the surface. (Note that this equilibrium response is different from the equilibrium response of a nudging approach, where a background flow is maintained. For the transient response we expect a similar behavior of both approaches.)"

b) more clearly communicate the "triviality" of the equilibrium response and focus more on the physically more interesting transient response and the UB2_unius experiment (line 68-70, 156-158):

"For the mechanism denial experiment a mean flow over the surface is maintained by including the influence of the mean wind only in the surface enthalpy equation but not in the surface momentum equation."

"The temporal evolution of the propagation speed demonstrates that the spin-down of the propagation speed occurs over a week whose time scale is longer than the convective adjustment time scale, which is in the order of hours, and the convective cluster settles around two weeks after it begins to propagate."

**2. The discussion focus on the surface fluxes, but the convection structure change from UB0 to UB4 is not discussed. The only clue given in this manuscript is the cloud top height in Fig.5. Based on my eye measurement, the horizontal scale of the convection in UB0 is around 100 km. But the scale shrinks to 50 km in UB2 and maybe 20 km in UB4 at the end of the simulation. It raises an interesting question: Can we still call convection in UB2 and UB4 aggregated convection (i.e., convection sustained by its circulation). The change in convective structure might also explain why further increase the wind speed, the convection disaggregated.**

We appreciate that the reviewer mentions this point. We have included a new figure 1 in the revised manuscript to display the horizontal structure of the convective cluster on the last day of the simulation period:

[Figure]

Figure 1 Daily average precipitable water on day 19. Black contours indicate where precipitable water is equal to 58 kg/m2

"Figure 1 illustrates the daily average spatial pattern of the convective cluster on the last day in the experiments. All simulations show that the quasi-circular pattern of the convective cluster lasts until the end of the simulation period, and the horizontal scale of the cluster size is comparable among all simulations, although the spatial variability of precipitable water is weak for UB4 compared to the other experiments. The standard deviation of the daily average precipitable water on the last simulation day is 14.2, 12.1 and 10.4 kg/m2 for UB0, UB2 and UB4, respectively. This standard deviation varies in time and, e.g., is as low as 10.9 kg/m2 on day 6 in the control case UB0. The domain mean precipitable water on the last day increases with increasing Ub, having the daily mean value of 26.5, 30.4 and 34.3 kg/m2 for UB0, UB2 and UB4. The larger domain mean precipitable water with increasing Ub might be associated with our simulation setup of a propagating cluster in double periodic boundary condition which results in nine full transits through the domain in case of UB4

(Matheou et al., 2011). Despite this artifact, the convective cluster remains organized over the simulation period in all experiments and we expect this difference to play a minor role in the following analysis."

We do not find the horizontal scale shrinks with increasing Uabs. In the Hovmöller diagram of the cloud top height one can easily be misled when estimating the horizontal scale because of the slanted pattern. In a new Fig. 1 in the revised manuscript, you can see the structure of the convective cluster for all experiments. In terms of daily average precipitable water convection is still aggregated for UB2 and UB4, although the spatial variability of precipitable water for UB4 is relatively weak compared to UB0, UB2 and UB2_unius. We performed UB4 for another 10 days, i.e., up to day 30, to confirm that self-aggregation maintains, and the convection is indeed aggregated until day 30. In contrast, a simulation with imposed mean wind of 5m/s, which is not discussed in the manuscript, begins to disaggregate convection around 5 days after imposing the mean wind.

**Minor comments:**
**In Fig. 1 The Uabs evolution for UB0, UB2, and UB4 are quite different during t=0 to t=5 days. The UB0, which should be very close to no imposed wind, has no fluctuation within first 5 days but fluctuate strongly afterward. This is similar to UB2 but with higher Uabs; on the other hand, the Uabs in UB4 decreases linearly within the first 5 days suggest the active WISHE feedbacks proposed by the authors. Do you have any explanation for this?**

When applying a broader running average (e.g., 72 hours) than what we used in Fig.2 in revised the manuscript (24 hours), UB2 and UB4 show a near-linear decrease of Uabs in the first 5 days. UB0 is the simulation with no imposed wind as a reference. The drop of Uabs for UB2 is associated with the fluctuation of the cluster shape. The cluster experiences expansion and contraction in all direction, but not necessarily uniformly in all directions at the same time. For example, when the cluster expands in y-direction, the PW-weighted center moves slightly away from the previous time step (in terms of x-direction as we present the propagation speed in x-direction in which we impose the mean flow) and when it contracts back, the PW-weighted center moves farther away from the previous time step. We have given more detail about the Uabs fluctuation (line 149-152 in the revised manuscript):

> "Since the cluster is formed by a group of individual convective cells, the shape of cluster is not firmly fixed. The cluster expands and contracts in time (though not necessarily in all directions at the same time, see the daily PW for UB2 in Fig. 1) and sometimes smaller convective cells emerge outside the main cluster (see the cloud top height for UB0 in Fig. 6)."

**Fig. 2 Quantities are averaged over 5 days and 10 km. What does 10 km mean?**

To get a smoother signal, we averaged quantities over 10 km in r-direction, not applying a running average. This detail has been updated in Fig. 3 and Fig. 4 in the revised manuscript.

**Line 48 a grid spacing of 75m at the first model level. Add up to XXX m near the model top.**

This has been included in line 53-54 in the revised manuscript:

> "The 63 vertical grid levels are stretched, starting from a grid spacing of 75 m at the first model level up to 1367 m near the model top."

**Line 89 day 22 doesn't make sense here. Do you mean 22 days after restart or 2 days after restart?**
**Line 91 same problem as line 89.**

We have clarified this (Sect. 2.2 in the revised manuscript). To check whether or not the disabled dynamic feedback affects the aggregation, we restart a simulation with suppressed dynamic feedback from day 22 of the analyzed simulations equivalent to -4 day of the spin-up simulation without a mean flow. This is included this in line 115-117 in the revised manuscript:

> "For the remainder of the study we refer to the simulation day, where we begin to impose the background wind, as day 0 (day 26 above). For example, the time when we restart the denial experiment without mean wind (day 22) would be equivalent to day −4, and the time when the mean wind is introduced to be imposed to the denial experiment (day 26) is day 0 from now on."

**Fig 2 and others, it's better to put figure legend to all the figures.**

Figure legend has been added to Fig. 3, Fig. 4 and Fig. 5 in the revised manuscript.

---

## Author Comment (AC2) · 21 Jan 2021

Dear Referee 2,

we thank the reviewer for the valuable comments and suggestions. We have carefully taken into account the comments and have reflected most of the suggestions in the revised manuscript. Here we repeat your comments in bold and write our responses in normal font.

**General Comments: This work uses the framework of a radiative convective equilibrium (RCE) experiment to investigate the effect of the wind-induced surface heat exchange (WISHE) mechanism on the moving speed of the active region of convection. By this unique approach, the authors found that the effect of WISHE to slow down the movement is very weak. Specifically, the quantification that the deceleration effect of WISHE is about 5% is valuable. The idea of the Galilean transformation to avoid difficulty in the discretization of the nonlinear term is interesting. Unfortunately, however, I cannot help recommending REJECTION of this work due to the inappropriate experimental configuration at this time. I am wondering if the authors really intended to include the momentum exchange between the surface and the atmosphere. As shown in this work, the surface drag acts to force the atmosphere to move with the ground, and the distinction between the transient response and the quasi-steady response becomes unclear. In my understanding, there is no positive reason to analyze the transient response. It seems that the authors already noticed this problem, and thus, they had performed the UB2_unius experiment. Would it be so difficult to include influence of the mean wind only in the surface enthalpy equation, but not in the surface momentum equation? One way may be to use two independent u_h values: one is the same as eq. (2) and the other is without u_b. If the coding is complicating and/or the computational cost is huge, analyzing the transition stage of the system may be acceptable. I think, however, that this is not the case. It would be desirable to revise the experimental settings and redo the whole experiments. After that improvements, the authors' arguments will be clearer, and it will make an essential contribution to the area of the RCE research.**

We agree with the reviewer in that it is interesting to perform an experiment that includes the influence of the mean wind only in the surface enthalpy equation, but not in the surface momentum equation. The reviewer is quite correct in his/her suggestion to decouple the momentum fluxes, but (and for the reasons the reviewer stated) we had already done this (UB_unius in the submitted manuscript). However, and in retrospect, this contribution was not presented in a particularly transparent manner. In the revised manuscript (line 65-70), we have substantially rewritten the manuscript to bring this issue more to the forefront.

> "In the long run with a mean flow the surface transports its signal through the atmosphere, until the whole column is in balance again and stagnant compared to the surface. (Note that this equilibrium response is different from the equilibrium response of a nudging approach, where a background flow is maintained. For the transient response we expect a similar behavior of both approaches.) For the mechanism denial experiment a mean flow over the surface is maintained by including the influence of the mean wind only in the surface enthalpy equation but not in the surface momentum equation."

We do think that in addition to the UB2_unius experiment it is not unreasonable to also learn from the transient response of UB2 and UB4, which include the influence of the momentum equation. The time scale of convective adjustment is in the order of hours, and in the simulations the convective cluster continues to propagate for several days during the transient phase. This time scale is a lot larger than that for the convective adjustment. We have included this in line 156-158 in the revised manuscript:

> "The temporal evolution of the propagation speed demonstrates that the spin-down of the propagation speed occurs over a week whose time scale is longer than the convective adjustment time scale, which is in the order of hours, and the convective cluster settles around two weeks after it begins to propagate."

Furthermore, we think it is informative to analyze the transient response is to better understand the role of the asymmetry of the surface momentum flux in slowing down the propagation of the convective cluster in our simulations. For these reasons, we think it is worth investigating both types of simulations in the manuscript.

**Other Comments: 1. Abstract: "phenomenon found in. . ." may be replaced with "phenomenon seen in. . ."**

We have replaced 'found' with 'seen' on line 1 in the revised manuscript:
> "Convective self-aggregation is an atmospheric phenomenon seen in numerical simulations in a radiative convective equilibrium framework of which configuration captures the main characteristics of the real-world convection in the deep tropics."

**2. It is desirable to review more specifically about the resemblance between the self-aggregation in RCE and the MJO. Since the pure RCE lacks the vertical wind shear, the resemblance between the MJO and the RCE is questionable.**

We do not intend to look at the MJO directly but see our study as a step to better understand how asymmetries in surface fluxes affect aggregation. One of the processes that might be important for the MJO is WISHE. But, if WISHE is important for the MJO, it is so in a different way than how we think about it in this study.

We have clarified this in the revised manuscript in line 32-33:
> " As a step, we focus on how asymmetries in the surface flux, in response to a mean flow, affect the propagation of a convective cluster in RCE."

**3. Note the number of vertical levels in section 2.**

The number of vertical levels has been added in line 54-55 in the revised manuscript:
> "The 63 vertical grid levels are stretched, starting from a grid spacing of 75 m at the first model level up to 1367 m near the model top."

---

## Author Response (AR2)

Dear Editor,

we would like to express our sincere thanks for your careful handling of our manuscript. We carefully considered your thoughtful recommendations to revise the manuscript. In the revised version of the manuscript, we now discuss the limitations of our setup in detail and we made an effort to clearly communicate the gap between our simplified setup and real-world tropical convection. Please find the details of our modifications to the manuscript along with our response to reviewer 2.

Sincerely

Dear Referee 2,

we thank the reviewer for the comments and suggestions. Here we repeat your comments in bold and write our response in normal font.

**I was disappointed when I read the revised manuscript and the authors response because it was obvious that the authors had not considered and responded to my comments seriously. For that reason, I considered recommending REJECTION again. As I said before, there are two major problems. One is that the emphasis on the relationship between this study and MJO is quite misleading. It is permissible to speak roughly about the implication of RCE on MJO. But WISHE is a controversial part of the MJO mechanism, and we should refrain from rashly confirming or denying its importance through studies that are not directly related to. The spatial scale of the organized system is about O(100km) in this study, and only the influence of WISHE on the mesoscale systems, which are in the subscale of the MJO, can be discussed.**

We would like to assure the reviewer that we took their comments seriously and are sorry if we failed to communicate that. We agree that the gap between this study and MJO was not communicated clearly in the manuscript. The potential relation between self-aggregation and the MJO (as described in the previous manuscript) was the motivation for our study, and we believe how this line of thinking led us to study a much simpler problem is important to acknowledge. However, we agree that it would be inappropriate to give the impression that our results end up being informative of our understanding of the MJO. Indeed, difficulties associated with our framework highlighted deficiencies in our understanding on a more basic level, in terms of attempting to decouple thermodynamic fluxes from those that alter the mean flow. In the revised manuscript we made an effort to avoid any ambiguity and clarified the gap between the motivation and focus of our study (line 12-14, 24-25, 34-35, 252-255).

> "In this manuscript we explore the simplest possible configuration that allows the interaction of a convective cluster with a mean flow. This is motivated by a desire to better understand processes influencing the propagation of organized deep convection in the tropics."

> "This leads us to the more basic question of how convective self-aggregation responds to the imposition of a mean flow."

> "This line of thinking leads us to attempt to study a much simpler problem, which is how convective-self aggregation responds to the imposition of a background mean flow."

> "While the problem we study is probably too simple to meaningfully inform our understanding of much more complex and larger scale processes like the MJO, it does highlight how a consideration of surface thermodynamic fluxes alone has only a small influence on the propagation of the convective cluster, and how considering these fluxes in isolation of the associated fluxes of momentum, distorts our understanding of the response to the asymmetry imposed by the mean winds."

**The other is that I am not convinced with the merit to analyze the transient response of the RCE system. As we see in Fig. 6b and c, the organized system moves in the x-direction of the reference coordinate, and this ruins the interesting Galilean transformation approach (to avoid the numerical difficulty in the advection term). As it is suggested by the different evolution of UB2 and UB4 in Fig. 2 (above), the "transient" responses (day0-5) of UB2 and UB4 may not be the same. Uabs of UB4 decreases nearly exponentially, but Uabs of UB2 is nearly constant during day0-5. In addition, it is clear from Fig. 2b (below) that UB4 is still in the transient phase for day15-20.**

We are aware of the limitation of our simulation setup and attempt to make this clear for the reader. Nevertheless, we think our results are worth an interpretation and are willing to share what we have learned throughout the study. We have included this in the experiment setup and conclusion section (line 91-95, 260-264, 268-270).

> "In retrospect, this modification ends up being effective only to a limited extent, as the advantage of a Galilean transformation to avoid numerical errors from advection is lost when the convective cluster start to move through the grid boxes. For futures studies that aim to study the interaction of convective self-aggregation with a mean flow, mechanisms for maintaining the mean flow must be included (e.g. a nudging of a large-scale flow), which couples the thermodynamic questions we had wished to study to dynamical ones."

> "A Galilean transformation can have the advantage of avoiding numerical artifacts of advection. The benefit of the approach, however, ends up being true only to a limited extent, as the convective system start to propagate through the model grid in our study. Nevertheless, the simulations show that the convective system maintains its thermodynamic structure until the end of the simulation period when $u_b \leq 4 \text{ ms}^{-1}$. For future studies, we recommend considering the momentum flux response to a large-scale motion by including a physical mechanism for maintaining a mean flow.

> The simplicity of our framework and the difficulties encounter in the setup of the simulations prevent direct inferences from our study for real-world propagating deep convection, let alone the MJO."

> "Nonetheless, the basic questions it highlights — such as the role of surface momentum fluxes in WISHE-like mechanisms — are likely to be fruitful avenues to explore when pursuing understanding of more complex phenomena."

To avoid overloading the figures, we display simulations with $u_b$ of $0, 2$ and $4 \text{ ms}^{-1}$ in the manuscript, although we have run several simulations with different wind speeds of $u_b$ and with ensemble perturbation (figure 1 in this document). There is some variability in the estimate of $u_{abs}$ but this set of simulations confirms that the decrease in $u_{abs}$ in the first days is systematic with increasing $u_b$. We have seen the systematic change in the $\theta_e$ flux at the surface and $F_m$ among the simulations, including Ub1, UB3 and ensemble runs as in Fig 3 in the submitted manuscript. We see the response to the imposed wind is consistent in the

transient phase when ub $\leq 4\ \mathrm{ms}^{-1}$. We additionally ran UB4 until day 30 which confirms that the simulation also reaches a quasi-equilibrium state. $u_{abs}$ for UB4 fluctuates around zero in the additional simulation days (day 20-30). The estimated propagation speed for UB4 (day 15-19) is 0.29 $\mathrm{ms}^{-1}$ with higher fluctuations around zero compared to UB0 and UB2. Since $u_{abs}$ converges to zero from day 15, we did not extend the simulation period for the other experiments. Thus, in the manuscript we decided to present the results using the period up to day 20 but this information has been added in the revised manuscript (line 155-158):

> "Additional simulations with $u_b$ of 1 and 3 $\mathrm{ms}^{-1}$ show agreement in that the propagation speed decreases in the first few days and eventually the propagation speed converges to zero (not shown). Additional simulation days for UB4 (until day 30) corroborate that UB4 reaches a quasi-equilibrium state (not shown)."

[Figure]

*Figure 1 Temporal evolution of $u_{abs}$ in the x-direction as in Fig 2 in the revised manuscript. UB1 and UB3 represent the simulations with an imposed wind flow of 1 and 3 $ms^{-1}$, respectively. UB3_ens and UB3_ens2 are the ensemble runs for UB3 where we slightly changed the maximum time step to 14 and 13s, respectively. (The maximum time step is set to 15s for all simulations in the revised manuscript.)*

**So, I cannot recommend publication of this manuscript in its current form, but obviously, that is just my opinion. I think that addressing the second point is particularly difficult because they have to redo all of the experiment and reinterpret it. So, at the very least, I would like to request that they rewrite Introduction and stop trying to force this study to be meaningful in relation to the MJO.**

We have revised the introduction to clearly communicate the gap between our study and the MJO (please see our answer to the first comment). The resemblance of the MJO and self-aggregation found in the previous studies, however, was the sincere motivation of our study and acknowledging this seems fair (as discussed earlier). The simplified framework and the difficulties encounter in the setup of the simulations make the results less relevant to the MJO than we had originally hoped. To account for this, we now critically discuss the

limitations of the setup in Section 2.1 and the Conclusion (please see our answer to the second comment). Also, we have adapted the abstract to clarify the objective of the study in the beginning (line 3-4):

> "We impose a background mean wind flow on convection-permitting simulations through the surface flux calculation in an effort to understand how the asymmetry imposed by a mean wind influences the propagation of convection."

---

## Author Response (AR3)

Dear Editor,

thank you for the thoughtful and considerate comments and suggestions on the manuscript. We have revised our manuscript. Here we repeat your comments in bold and write our responses in normal font.

**You have made further adjustments to the paper in response to the comments one Referee 2 and I think that the fair option is now to proceed very quickly to accept the paper for publication in ACP.**

**On the basis of reading through the latest version of the paper myself I make the comments below -- please consider making further modest changes in response to those comments and provide a final version of the paper -- which I will then be pleased to accept.**

**abstract: 'aggregation' is mentioned in the first sentence of the abstract, but the term is not used again in the rest of the abstract. A reader might wonder whether the first sentence has any relevance to the rest of the abstract. I think that you are interested in the effect of a mean flow on the aggregated structure -- saying something like 'propagation of aggregated structures in convection' in the second sentence would help the reader see the link.**

**abstract l9: 'retards the propagation of the convection' would be clearer than 'retards the convection'. Incidentally this last line of the abstract seems an important conclusion, but '5% of the mean wind' may be misleading -- you don't actually know whether the effect is a fixed percentage of the mean wind (as the mean wind varies).**

We appreciate Editor's suggestions to clarify the content in the abstract. We have modified the words as suggested.

**l109: 'mechanism denial experiments' -- but in fact there is only one experiment.**

This has been corrected to 'a mechanism denial experiment' (line 112).

**l145:**
**'In the case of WISHE, the convective cluster moves against the mean wind (e.g.,urel <0ms−1).' -- what you actually mean here I think, is 'If the hypotheticated effect of WISHE was realised then the convective cluster would move against the mean wind'**

This has been updated in line 150-151 in the revised manuscript.

**l249: 'Even in the simulation with the dynamic feedback removed and the WISHE-induced asymmetry in surface fluxes preserved, the effect on the propagation of convective clusters is small.' -- why 'Even'? Isn't the point that in this case the effect is sustained, but weak by some measure?**

Yes, this is the point that the effect is sustained but weak by some measure. The misleading adverb 'Even' has been removed in line 257 in the revised manuscript.

**Two further comments -- which you may or may not wish to take into account in producing the final version of the paper. I am essentially setting out my own understanding of the paper -- if I have misunderstood something then you may wish to consider whether other readers may also have similar misunderstandings -- and alter the text a bit to avoid this. (Of you may consider that my misunderstanding unlikely to be shared by most readers.)**

We would like to express our appreciation that the Editor shares this with us. This is important to clearly deliver what we learned from our study and to communicate with readers. Therefore, we have made changes to address the issues.

**1) My own interpretation of your findings is that, whilst the effect of the mean flow might according to some have a significant effect on the propagation because of the associated difference in surface fluxes upstream and downstream of the aggregated system. You are intending to identify this effect by considering u_rel and finding that it is negative (i.e. in the opposite sense to the imposed mean flow. What you find is that u_rel is certainly negative, but a large part of the explanation for this is that the upstream/downstream different cannot be sustained close to the surface inding is that this effect cannot be sustained because the momentum exchange with the surface limits the upstream/downstream difference -- the low-level mean flow is quickly reduced in magnitude. Of course this does not rule out a sustained effect in a different system where there is active dynamical driving of the low-level flow.**

> (line 171-172) "In the following sections we examine if the tendency towards stationarity is a consequence of WISHE-like symmetries by means of an upstream/downstream difference."

> (line 221-223) "The upstream/downstream difference cannot be sustained close to the surface because the momentum exchange limits it in our simulations. This does not rule out a sustained effect in a different system where there is an active dynamical driving of a low-level flow."

**2) Personally every time I looked at this paper I had difficulty relearning what system it was exactly that you were studying. I think that you are considering a system in which, from the point of view of an observer fixed relative to the Earth's surface, at t=0 the air velocity at all levels is u_b, and you choose to observe/simulate the system whilst moving at speed u_b.**

**When the effect of the surface is transferred to the atmosphere above by a momentum flux, the velocity in the atmosphere naturally reduces towards that of the surface, i.e. in your frame of reference the velocity in the atmosphere naturally adjusts towards the value -u_b. When there is no momentum flux from the surface then the velocity in the atmosphere remains at u_b, i.e. in your frame of reference it remains at zero.**

**If you could make a simple statement saying something like that then it would help a lot -- the reader would not have to worry about which physical effects were included in your formulation -- equations (1) and (2) -- and which were not.**

We tried to explain how the system works as easy as possible by introducing the conveyor belt metaphor but acknowledge that the description remains unclear. To clarify this, we have made changes in the revised manuscript:

> (line 71-72) "In the long run when the effect of the modified surface fluxes is transferred to the atmosphere above by the momentum flux, the velocity in the atmosphere naturally reduces towards that of the surface."

> (line 91-92) "From the point of view of an observer fixed relative to the moving surface, at t = 0 the air velocity at all levels is $u_b$, and the model framework is also moving at speed $u_b$."

> (line 95) "… when the air and the convective cluster start to move …"

> (line 158-159) "(At this point the convective cluster appears stationary to the observer.)"